# Biological Characterization of Polymeric Matrix and Graphene Oxide Biocomposites Filaments for Biomedical Implant Applications: A Preliminary Report

**DOI:** 10.3390/polym13193382

**Published:** 2021-09-30

**Authors:** Thamires Santos Silva, Marcelo Melo Soares, Ana Claudia Oliveira Carreira, Gustavo de Sá Schiavo Matias, Carolina Coming Tegon, Marcos Massi, Andressa de Aguiar Oliveira, Leandro Norberto da Silva Júnior, Hianka Jasmyne Costa de Carvalho, Gustavo Henrique Doná Rodrigues Almeida, Michelle Silva Araujo, Paula Fratini, Maria Angelica Miglino

**Affiliations:** 1Department of Surgery, School of Veterinary Medicine and Animal Science, University of São Paulo, São Paulo 05508-270, Brazil; thamiresssilva@usp.br (T.S.S.); ancoc@iq.usp.br (A.C.O.C.); gustavoschiavo@usp.br (G.d.S.S.M.); silvajunior@usp.br (L.N.d.S.J.); hiankacarvalho@usp.br (H.J.C.d.C.); gustavohdra@usp.br (G.H.D.R.A.); msa.vet@usp.br (M.S.A.); paulafratini10@gmail.com (P.F.); 2Department of Materials Engineering, Mackgraph Institute, Mackenzie Presbyterian University, São Paulo 01302-907, Brazil; marcelomelo61@gmail.com (M.M.S.); caroltegon@gmail.com (C.C.T.); marcos.massi@mackenzie.br (M.M.); andressa_120@hotmail.com (A.d.A.O.)

**Keywords:** biocompatibility, mesenchymal stem cells, tissue engineering, nanomaterials, carbon nanosheets, poly (L-lactic acid)

## Abstract

Carbon nanostructures application, such as graphene (Gr) and graphene oxide (GO), provides suitable efforts for new material acquirement in biomedical areas. By aiming to combine the unique physicochemical properties of GO to Poly L-lactic acid (PLLA), PLLA-GO filaments were produced and characterized by X-ray diffraction (XRD). The in vivo biocompatibility of these nanocomposites was performed by subcutaneous and intramuscular implantation in adult Wistar rats. Evaluation of the implantation inflammatory response (21 days) and mesenchymal stem cells (MSCs) with PLLA-GO took place in culture for 7 days. Through XRD, new crystallographic planes were formed by mixing GO with PLLA (PLLA-GO). Using macroscopic analysis, GO implanted in the subcutaneous region showed particles’ organization, forming a structure similar to a ribbon, without tissue invasion. Histologically, no tissue architecture changes were observed, and PLLA-GO cell adhesion was demonstrated by scanning electron microscopy (SEM). Finally, PLLA-GO nanocomposites showed promising results due to the in vivo biocompatibility test, which demonstrated effective integration and absence of inflammation after 21 days of implantation. These results indicate the future use of PLLA-GO nanocomposites as a new effort for tissue engineering (TE) application, although further analysis is required to evaluate their proliferative capacity and viability.

## 1. Introduction

New efforts applied to tissue engineering (TE) aim to replace or at least repair organs and tissues through transplants (allo, auto and xenotransplantation) [1,2,3]. Among the TE developed approaches, the use of nanomaterials has shown promising results to solve challenges related to surgical clinics [4]. A promising approach for obtaining specific autologous tissues/organs is the 3D bioprinting technique, which offers a highly automated manufacturing (MA) platform for the fabrication of complex bioengineered constructs [5,6].

Recently, in order to improve these problems, carbon nanostructures such as graphene (Gr) and graphene oxide (GO) are providing new perspectives for TE. These can be applied in drugs and genes delivery of, as well as in the articular repair and bone defects [3,7,8,9,10] due to their physical characteristics such as mechanical strength, flexibility, elasticity [11], low density [12], structural support, self-lubricating properties and anti-wear due to its laminar structure which supports in the surface lubrication. Furthermore, these can be applied to mediate cell proliferation, differentiation and migration and improve the bone repair effect [7,13,14,15,16]. Additionally, GO plays an important role in stimulating cell functions and guiding tissue regeneration, as presented by Du et al. [7], comparatively showing the osteogenic capacity of treated multiple-wall carbon nanotubes (MCNTs) and the inorganic component of natural bone. Although there was no significant difference in the amount of cell adhesion between groups, fixation strength and cell proliferation were better, in addition to the fact that these materials activate signaling pathways related to bone induction. 

Gr has prospective application in various fields of tissue engineering [17,18,19,20,21] since its structure consists of a monolayer of carbon atoms, with a basic hexagonal compacted structure, one-dimensional and three-dimensional (graphite) [21]. This structure has specialized qualities related to other metals, such as mechanical strength, high electrical, thermal and magnetic properties. On the other hand, GO is a graphene-derived nanosheet structured by epoxy, hydroxyl and carboxyl groups on its surface that make it more hydrophilic. These groups also promote biological interactions with nanosheets allowing their use for neural regeneration in tissue engineering [22] and providing a compatible environment for chondrogenic cell proliferation [3].

Moreover, studies have shown better capacity of GO when linked with mechanical and electrical properties of naked materials. GO association with other materials, such as biocompatible copolymer and biodegradable polyester (BP) acquired from a microbial source, reveals an increase in its piezoelectric, mechanical and tensile strength properties of combined BP+GO compared to PB by own [10]. As a result, the GO association demonstrates a potential use for cartilage repair, showing that scaffolds reinforced with GO reveal greater adhesion and cell proliferation. Additionally, Trucco et al. [3] demonstrated that GO associated with gellan gum (GG) and hydrogel of poly diacrylate (PEGDA) in order to mimic the mechanical and lubricating characteristics of superficial and deep articular cartilage. 

Poly (L-lactic acid) (PLLA), in turn, is highly used as a biocompatible polymer in medical devices, commonly applied in osteosynthesis bioengineering due to its greater commercial potential as a versatile biodegradable plastic, its thermoplastic processability, low price and great mechanical properties [23], and for its biodegradability and biocompatibility. These qualities added to its use in the device’s design for bone fixation and controlled drug release [24,25,26].

Finally, this manuscript aimed to characterize production of the filaments by PLLA and GO nanocomposites associations, verifying the capacity of MSCs to adhere to nanocomposites after 7 days in vitro. Thus, the biocompatibility was tested in vivo through the implant nanocomposites in the subcutaneous and intramuscular of adult Wistar rats in a pilot study, hypothesizing that after 21 days, there will be no direct immune response at the implantation site.

## 2. Materials and Methods

### 2.1. Samples

This work was carried out in agreement with the rules of the Ethics Committee on Animal Use, CEUA/FMVZ-USP (Protocol number CEUA 9130071019/IB 4959310120).

### 2.2. Nanocomposites Characterization

Graphene oxide was produced from the chemical exfoliation of graphite (Nacional de Grafite Ltda^®^, São Paulo, Brazil) by the modified Hummer’s method [27]. A fraction of 0.2% (wt) of GO was first incorporated into the PLLA pellet (Evonick RESOMER^®^ L 210S), (PLLA-GO, Mackgrape Laboratory of Mackenzie University, São Paulo, Brazil) at the same extruder zone (Thermo Fisher Scientific Process no 11 (extruder standard screw), L/D 40, Waltham, MA, USA), in a temperature profile (Table 1) producing 1.75 mm thickness filaments. No plasticizer was used. The mechanical energy was not calculated.

The nanocomposites produced were characterized according to their crystallographic structure and arrangement by X-ray diffraction (XRD) measurements. The samples were cold pressed and placed in the diffractometer sample holder (Rigaku Miniflex II), equipped with copper filament (CuKα) at room temperature, with an angular range of 5° to 60°, and a rate of 2°/min.

### 2.3. Nanocomposites In Vivo Biocompatibility Test

Nine adult Wistar rats (five females and four males, body mass ranging: 350–550 g) were used for the biocompatibility test of nanocomposites and their immune response analysis, implanted subcutaneously in the posterior limbs (bilateral region of the flank) and the gastrocnemius muscle. After nanocomposite implant surgery, opioid tramadol analgesia was performed using a 20 mg/Kg (I.M) dose, diluted 2.5 times in ringer’s lactate solution. The animals were placed in numbered cages (1 animal/cage) containing sterile wood shavings, enriched environmental and appropriate rooms with controlled temperature (20–22 °C), and a light/dark cycle. The animals’ diet consisted of commercial sterile pelleted feed and autoclaved filtered water available ad libitum. Animals were evaluated daily.

After 21 days, the animals were euthanized by administering 30 mg/Kg (I.P.) of Xylazine and 300 mg/Kg (I.P.) of Ketamine. Euthanasia was confirmed due to the heart and respiratory rate absence. The implantation regions were photo-documented, the fragments were collected and stored at −80 °C until histopathological and Scanning Electron Microscopy (SEM) analysis was processed.

### 2.4. GO Nanocomposite Subcutaneous Implant

Four (female rats) were used to test implantation of GO powder (2 g), and one animal (female rat) was used as a control. The nanocomposites were placed in 60 mm plates, washed 3 times with Phosphate Buffered Saline 1X ((PBS) 136.9 mM NaCl, 26.8 mM KCl, 14.7 mM KH_2_PO_4_ and 8 mM Na_2_HPO_4_, pH 7, 2, LGC Bio), sterilized under UV light in laminar flow (BIO SEG 12, Class II, Type AI) for 15 min. The animals were sedated with 2.5 mg/Kg (IP) Acepromazine and then submitted to anesthetic induction with 10 mg/Kg (IP) of Xylazine and 100 mg/Kg (IP) of Ketamine, diluted in saline solution at a ratio of 1: 1:3 (*v*/*v*/*v*). After local antisepsis, a unique incision (1.5 cm) was performed. Then, the skin was divulged for surgical tissue deepening and insertion of the GO powder. After implantation of the material, sutures were performed in a simple pattern interrupted with thread (Shalon^®^, nylon 5–0). Following the experimental period, the animals were photo-documented and the samples collected for analysis. 

### 2.5. PLLA-GO Nanocomposite Intramuscular Implant

Four animals (male) were used for bilateral implantation of PLLA-GO nanocomposites in the gastrocnemius muscle. The nanocomposite was previously sterilized in an autoclave (Bioex) for 15 min, handled in laminar flow (BIO SEG 12, Class II, Type AI) and quickly washed in 1X PBS. Subsequently, using local antisepsis (70% alcohol), an incision was performed (1.5 cm) dorsally to the gastrocnemius muscle to expose the muscle fascia. Then, deepening was performed to expose the epimysium and, subsequently, the perimysium. Following muscle dissection, the PLLA-GO (3 mm × 1 mm, 0.017 g) was implanted, and sutures were performed (Shalon^®^, Catgut simples 3–0) in the muscle fascia region (Shalon^®^, nylon 5–0) in the skin. Following the experimental period, the animals were photo-documented, and the samples collected for analysis. 

### 2.6. Histological Analysis

The implantation test samples (GO, PLLA-GO) were fixed in 4% paraformaldehyde (PFA) for 48 h. Then, the samples were dehydrated in an increasing series of alcohol, 70%, 80%, 90% and 100% for 30 min each. Subsequently, the samples were immersed in Methyl methacrylate (C_5_H_8_O_2_) for 15 days, and later cut manually with a goldsmith saw (AmploTech). The acrylic slides were stained using the hematoxylin and eosin (H&E) and Masson’s Trichrome technique and analyzed under a light microscope (FV1000 Olympus IX91, Tokyo, Japan) of the Advanced Diagnostic Imaging Center—CADI-FMVZ/USP.

### 2.7. Culture of MSCs Associated with PLLA-GO Nanocomposites 

PLLA-GO fragments were thrice washed with PBS supplemented with 1% antibiotic (ATB, LGC- Penicillin-Streptomycin) for 5 min each, placed in 35 mm plates, washed with a splash of 70% alcohol and exposed to ultraviolet (UV) light for 15 min, and thrice washed with PBS again. Next, 5 × 10^4^ cells from canine dental pulp and canine adipose-tissue were used, as previously established by our group [28]. The cells were cultured using α-MEM medium (LGC), supplemented with 10% fetal bovine serum (SFB, LGC) and 1% ATB for 7 days at 37 °C and 5% CO_2_. Afterwards, the samples were PFA fixed for 72 h for further analysis by SEM.

### 2.8. Scanning Electron Microscopy (SEM)

The fragments (PLLA-GO + cells) were fixed in 4% PFA for 72 h. Following this, the fragments were dehydrated in increasing alcohols (70–100%), dried in a critical point machine (Leica EM CPD 300^®^) and glued with carbon tape in metallic (sputter coating) with golden metallizer (# K550 Emitech, Ashford, UK). Lastly, they were analyzed in a scanning electron microscope (LEO 435 VP^®^).

## 3. Results and Discussion

### 3.1. Nanocomposites’ Characterization

The processed biocomposite was evaluated by X-ray diffraction. As a result, it was possible to observe new crystallographic planes formation when graphene oxide was incorporated at an angle of 27° and 32° (Figure 1). This modification made the material amorphous to crystalline. The degree of crystallinity (CD) for 0.2% increased about 10% compared to the pure PLLA, resulting in the absolute number calculation by differential scanning calorimetry (DSC). This was carried out by calculating the CD of the pure PLLA, obtaining 31.6% and for 0.2%, obtaining 42.5%. 

New crystallographic peak formations could also be related to the formation of the -L and -D stereocomplex due to the temperature range and processing conditions performed. Additionally, GO acts as a nucleating agent, increasing the degree of crystallinity and the rate of crystallization, resulting in structure rearrangement. Similarly, research by Sun et al. [29] evaluated the graphene-poly oxide (lactide) nanocomposites synthesis and crystallization. They found that the formation of new crystallographic peaks occurred at the angles of 20.7° and 23.9°. The variation found in the angles was correlated with the GO presence and incorporated in the temperature processes used.

In addition, XRD evaluation shows displacements to the left and right, resulting in biocomposites with different crystallinity, also shown in thermal analysis results. Primarily, results showed that the further to the left, the greater the spacing between the planes. These results could indicate the interaction of the functional groups with the side groups, increasing the spacing of the sheets. The 200 and 110 planes also reflect the PLLA crystal structure, around the peak at 27° and 32°. This reveals their displacements according to the GO dispersion and concentration, being quite noticeable between the biocomposites according to the diffractogram, indicating a chains rearrangement [30]. 

Wang and Qiu [31] evaluated the PLLA-GO nanocomposites’ crystallization behavior from the amorphous state. They observed that the PLLA crystallization peak temperature changed to a low-temperature range in PLLA-GO nanocomposites following the increase in GO charge compared to pure PLLA at the same heating rate. Thus, it was possible to indicate that the non-isometric cold crystallization behavior of PLLA was improved by the GO presence.

Likewise, Sun and He [29] synthesized and characterized PLLA with GO-g-PDLA. The XRD studies showed that a stereocomplex crystal could be formed between PLLA and GO-g-PDLA due to the incorporation of GO nanofills, which in turn leads to a lower activation energy of stereocomplex crystallization and a higher crystallinity in solution casting samples, mainly due to the GO sheets heterogeneous nucleating effects. Otherwise, the crystallinity was low in cold crystallized samples due to the exfoliated GO sheets, indicating the reduction in chain mobility and hindered crystal growth.

### 3.2. In Vivo Analysis

After 21 days of GO subcutaneous implantation, the surgical wound scar was established (Figure 2a). Upon sectioning, it was observed that the occurrence of GO particles organized in ribbons formed, fixed to the subcutaneous tissue (Figure 2b). The healing process occurred as expected, without characteristic purulent alterations or pyogranulomatous inflammation and was a mainly granulomatous inflammatory process.

The GO biocompatibility is controversial in the literature [32], since this material, when tissue interaction occurs, adapts to its new situation by causing a local inflammatory response, acting both as bacteriostatic and increasing cell adhesion. This response occurs by releasing mediators that induce alterations in the phagocytes microcirculation and migration to the interstitium [33]. On the other hand, Gr does not act toxic in living organisms, being ideal for biological applications [34]. 

One of the greatest applications of these nanocomposites is in drug delivery [8]. With the aim of reaching specific sites efficiently and minimizing the possible side effects triggered by some drugs such as chemotherapeutics, composite associations were tested to evaluate the “intelligent bioavailability” properties of such drugs [8]. Associations between GO and other nanocomposites showed relevant properties such as in vitro cytocompatibility [35], high target site specificity with the decreasing of drugs and ectopic sites action [36] and multi-drug co-delivery capacity [37]. However, several studies showed limitations in demand for in vivo assays to assess the biocompatibility of these nanocomposites in complex organisms, such as immunogenicity parameters of these materials, organs and tissues integration, and their excretion and metabolization pathways [8]. Here, we prioritize animal models’ biomaterials application in order to obtain a more reliable tissue response, which would not be possible with an in vitro assay.

The intramuscularly implanted animals with PLLA-GO (Figure 3a–c) show good biomaterial integration to the adjacent tissue, without inflammatory infiltration, and with small reddish regions diffusely distributed over the tissue after 21 days implantation. It is likely that the reddish areas observed occurred due to the tissue fragment removal attached to the biomaterial, which caused a slight hemorrhage of the small vessels that irrigate this region. It is noteworthy that this hemorrhage is not visible on the material (Figure 3c) and, therefore, is not related to an inflammatory process triggered by its presence. Additionally, no inflammatory process signs were observed on the surgical wound, such as exudate, heat, swelling and redness [38]. Furthermore, the surgical wound promptly healed without any inaccuracies.

Studying the biochemical effects on different tissues of white shrimp (*Litopenaeus vannamei*) exposed by GO ingesting in the feed, no histological changes were observed in several organs, including muscle tissue [39]. These findings were similar as observed in the present research, although the material was implanted surgically. These results can be explained by the GO flat structural form, allowing these nanosheets to aggrege to cell membranes [40]. Once inside the cell, these materials can escape from cell compartments, migrate to the cytoplasm and translocate to the nucleus, and may also undergo oxidative degradation [41]. In the intramuscular test performed here, this did not occur since the size of the evaluated nanocomposites had integration with the tissues.

By histological analysis, in general, tissues presented similar characteristics. The subcutaneous tissue from the control animal showed well-defined layers of dense unmodeled connective tissue, showing elastic and collagen fibers, and it was also possible to observe the hair follicle preserved (Figure 4a). The animals that received the GO implant had a superficial layer of dense unmodeled connective tissue and a layer of adipose tissue surrounded by GO nanocomposite (Figure 4b). In the latter, the GO nanoparticles linked to form a ribbon, resulting in good tissue integration target to the subcutaneous layer, with a flexible characteristic. The same histological pattern was observed in samples stained with HE (Figure 4c,d).

Such results are similar to the analysis by Veetil et al. [12], which demonstrated that when merged into other structural materials, GO promotes structural reinforcement, giving new properties such as electrical conduction and directly supporting cell growth. Additionally, graphene and its derivatives increase the physical–mechanical properties of polymers, demonstrating that when functionalized, they cause high biocompatibility [42]. 

Otherwise, an important aspect for nanomaterials application in biomedicine is their relative cytotoxic effect on higher animal cells. A few nanomaterials, such as Ag, ZnO, TiO, and carbon nanotubes, indicate moderate to high levels of cytotoxicity against a variety of animal cells. However, the nanomaterials toxicity in some animals is highly specific to the cell type and depends on nanomaterial concentration [43]. 

The non-toxic and biocompatible behavior of several nanomaterials in relation to animal and human cells has been referred to in the literature [44]. Regarding GO, its toxic effect depends on several factors, including the route of administration, dose to be administered, method of synthesis and its physicochemical properties [45]. These factors directly influence and increase the complexity of comparisons between different studies about its toxicity [32].

Vuppaladadium et al. [46] researched GO biocompatibility when applied intravenously in mice. Doses of 0.1 mg and 0.25 mg did not exhibit cytotoxicity. However, at a dose of 0.4 mg, the animals presented chronic toxicity, with the death of 4/9 of them. Thus, it is possible to conclude that the nanomaterial size, added to the dose to a greater extent, was a determinant in the area and pattern of contact between nanomaterials and the cell membrane [47]. Particle size and surface area are essential parameters that play a significant role in the nanomaterials’ interaction. As a particle’s size decreases, its specific surface area increases, and the number of atoms becomes more considerable [48]. 

By histological evaluation of the tissue fragments from the control animal, it was possible to observe a well-evidenced muscle layer, and the elastic fibers were arranged in different directions (Figure 5a,c). The same histological pattern was observed in animals with implants, with no difference in the contact interface between PLLA-GO and adjacent muscle tissue (Figure 5b,d).

GO is a carbon nanosheet that has semiconducting characteristics due to its oxidation level [49]. This characteristic makes it an eligible candidate for use in tissue engineering since its excellent conductivity influences muscle tissue formation [50]. One of the unique behaviors of muscle tissue is its contraction activity due to electrical signals response [51]. Thus, the ability to conduct electrical signals induces its use as a conducting polymer biomaterial for muscle tissue engineering [52], carbon nanomaterials [53] and metallic nanomaterials [54].

By SEM, the ultrastructural and topographic visualization of the tissue surface was analyzed, as well as the microstructure of the materials after surgery and tissue integration. The animal control skin (Figure 6a) shows well-defined collagen fibers, adipose tissue and hair follicle. In animals submitted to GO implantation, it is possible to observe the material integration attached to the tissue while the subcutaneous tissue layers were preserved (Figure 6b). 

Likewise, studies carried out with GO, and other nanocomposites for orthopedic cement showed high levels of biocompatibility. Furthermore, the SEM analysis revealed that the structuring skin cells were well distributed on the material’s surface, demonstrating a particular characteristic morphological tissue view [55]. On the other hand, in animals implanted with PLLA-GO, it was possible to observe the successful material integration, preserving the muscle layer (Figure 7a) and the space surrounded by the polymer (Figure 7b).

### 3.3. In Vitro Analysis

The mesenchymal stem cells culture (MSCs) derived from canine dental pulp cells and adipose tissue associated with PLLA-GO (Figure 8a,b) showed cell adhesion and stretching in the biomaterial after 24 h. The findings observed here are similar to Unagolla et al. [56], which demonstrates changes in crystallinity caused by stereocomplexes. These findings indicate that stereocomplexes could cause favorable alterations in cell adhesion by forming functional groups containing oxygen and favoring interactions with hydrogen bonds. These results are similar to findings by Vlček et al. [57], which report electron cloud π (pi) integration, favoring hydrophilicity.

Studying the relationship of cell adhesion to two-dimensional carbon-based materials is necessary to increase GO applications in the most varied areas. Vlcek et al. [57] showed that by researching the shift kinetics of cancer cells adhered to the GO, proved strong adherence to this nanomaterial, and the presence of hydrophilic functional groups apparently increased the cell adhesion to the GO surface. Moreover, reports demonstrated that human fibroblast cells present apoptosis when exposed to medium culture containing high levels of GO; however, when cultivated in low concentration, they showed compliance with regard to their cell adhesion and parameters [58]. 

Several studies report PLLA activity related to cell adhesion [59,60]. In addition to cell adhesion proofing, Yanagida et al. [61] for instance, verified the response to PLLA tissue coated with hydroxyapatite nanocrystal (HAp). The researchers observed that cell adhesion was improved to the surface of PLLA/HAp using fibroblast cell lines. Furthermore, the inflammatory response was reduced to PLLA tissue compared to untreated, suggesting its use of these materials in the fields of orthopedic surgery and as a cell scaffold in TE.

## 4. Conclusions

Finally, by PLLA/GO anchoring to MSCs, it was verified that there was successful adhesion under the surface of the nanocomposites. However, further studies are required to evaluate their proliferative capacity and viability. Furthermore, the in vivo biocompatibility test demonstrated effective integration and absence of inflammation after 21 days of implantation, indicating its future use as a new application for tissue engineering. The first step has already been taken.

## Figures and Tables

**Figure 1 polymers-13-03382-f001:**
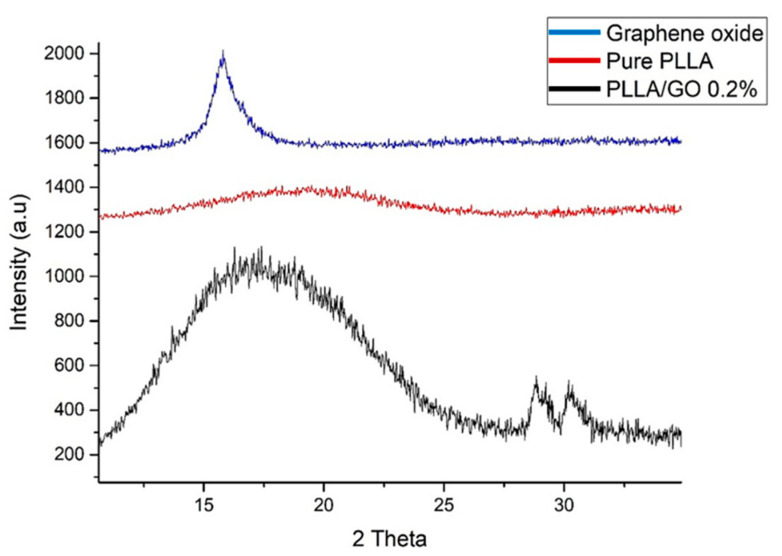
Pure PLLA, GO and processed biocomposite X-ray diffraction (XRD) analysis.

**Figure 2 polymers-13-03382-f002:**
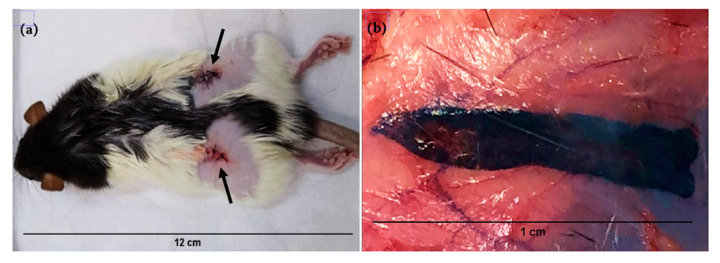
Surgical wound (arrows) after GO powder implantation in the subcutaneous tissue (**a**). 21 days evaluation of implanted biomaterial. GO ribbon agglutination form (**b**).

**Figure 3 polymers-13-03382-f003:**
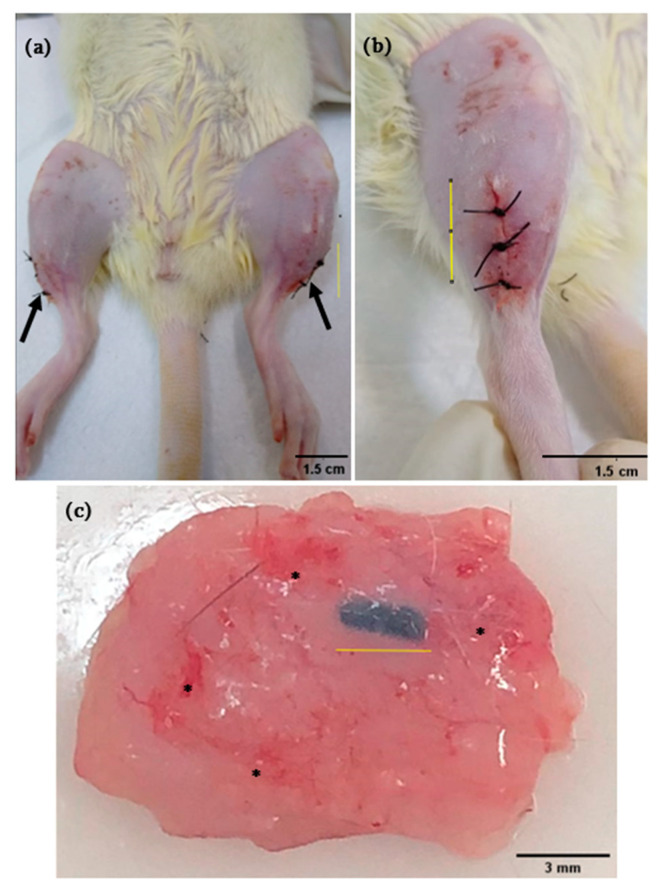
Ventral and right lateral view (**a**,**b**) of surgical wounds after bilateral PLLA-GO intramuscular implants (arrow); Muscle tissue fragment with the completely implant adhered, with small reddish areas randomly distributed (*) after 21 days (**c**).

**Figure 4 polymers-13-03382-f004:**
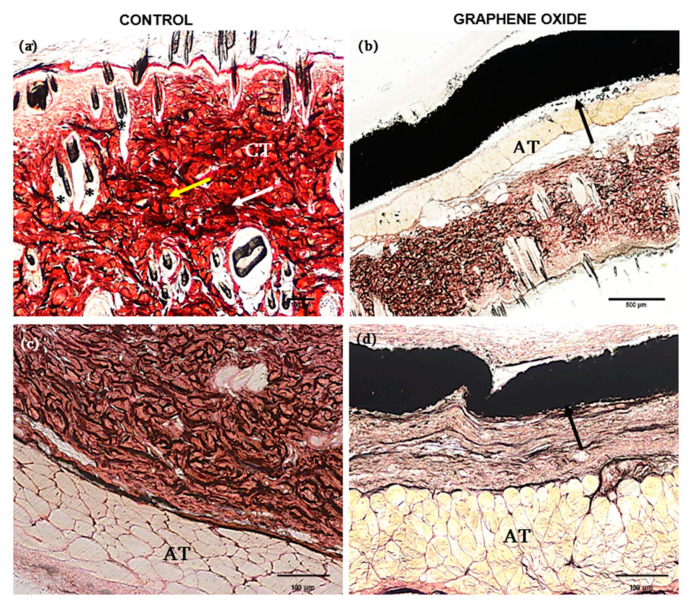
Skin histological section (control sample) stained with Masson’s Trichrome (**a**) and HE (**c**), respectively, showing the dense unmodeled connective tissue (CT), collagen fibers (yellow arrow), elastic fibers (white arrow) and hair follicle (*), scale bar: 100 µm (**a**); Skin histological section with GO e (black arrow) stained with Masson’s Trichrome, highlighting the dense unmodeled connective tissue (CT) and adipose tissue (AT), scale bar: 500 µm (**b**) and HE (**d**) respectively, scale bar: 100 µm.

**Figure 5 polymers-13-03382-f005:**
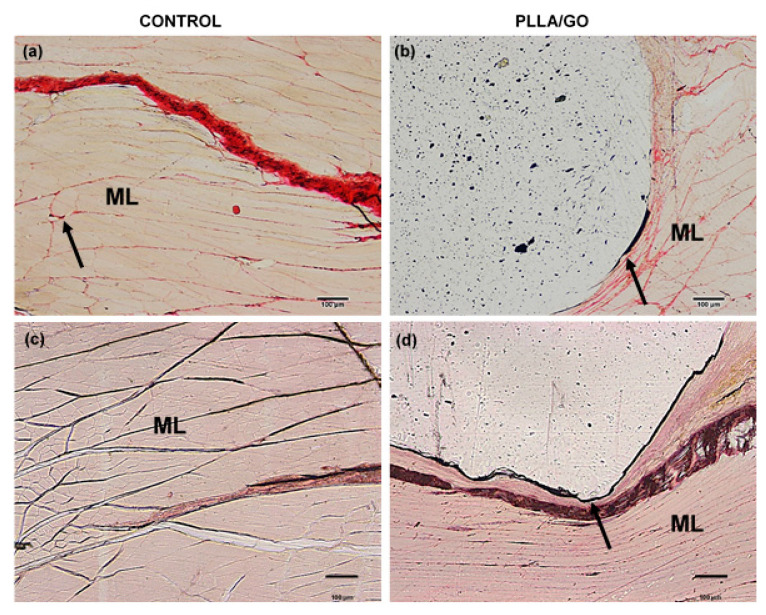
Gastrocnemius muscle histological section of control sample stained with Masson’s Trichrome and HE, highlighting the muscle layer (ML) and elastic fibers (arrow), bar: 100 µm (**a**,**c**), respectively; Gastrocnemius muscle with PLLA-GO histological section, stained with Masson’s Trichrome and HE, highlighting the muscle layer (ML) and the PLLA-GO insertion site indicated by (arrow), bar: 500 µm (**b**,**d**), respectively.

**Figure 6 polymers-13-03382-f006:**
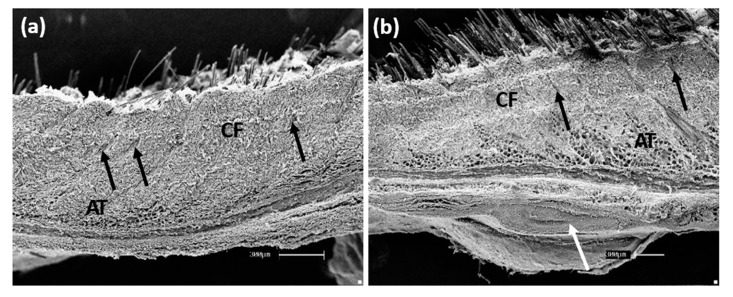
Scanning electron microscopy (control animal), with collagen fibers (CF), adipose tissue (AT) and hair follicles (arrows), bar: 300 µm, (**a**); Skin with GO indicating collagen fibers (CF), adipose tissue (AT), the hair follicle (black arrow) and GO layer (white arrow), scale bar: 300 µm, (**b**).

**Figure 7 polymers-13-03382-f007:**
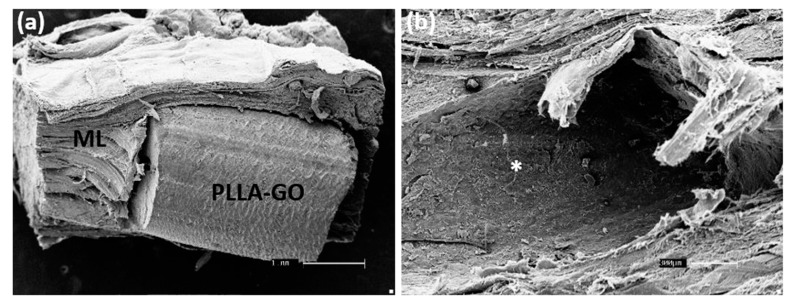
SEM of the muscle layer (ML) containing the polymer (PLLA-GO), bar: 1 µm, (**a**); focal local indicating the polymer space after removing (*), scale bar: 300 µm (**b**).

**Figure 8 polymers-13-03382-f008:**
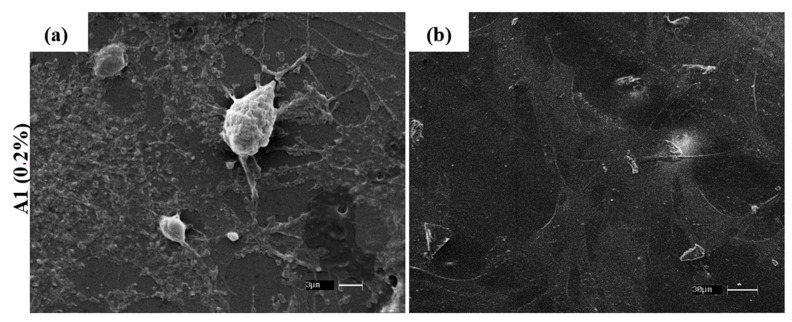
SEM analysis, showing the canine MSCs from dental pulp and adipose tissue attached to the nanocomposites (PLLA-GO) after 7 days culture. Bars: 3 µm (**a**) and 30 µm (**b**).

**Table 1 polymers-13-03382-t001:** Temperature zones profile.

DIE	Zone 8	Zone 7	Zone 6	Zone 5	Zone 4	Zone 3	Zone 2
220 °C	220 °C	215 °C	210 °C	210 °C	210 °C	210 °C	200 °C

## Data Availability

The data presented in this study are available on request from the corresponding author.

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
