# Peer review of "Biological Characterization of Polymeric Matrix and Graphene Oxide Biocomposites Filaments for Biomedical Implant Applications: A Preliminary Report"

_polymers, 2021, doi:10.3390/polym13193382_

Round 1

Reviewer 1 Report

Refer to the reviewer report 

Author Response

Dear Reviewer, 

Thank you!

Maria Angelica Miglino.

Reviewer 2 Report

This work presents a GO-PLLA nanobiocomposite that is biocompatible and could be a great advance in Tissue Engineering. It is novel and could be interesting for the readers of this journal. Nevertheless, the authors must be considered the following comments before its publication:

1) Define MSCs with its complete name the first time that you use it.

2) The groups of references in the introduction (i.e. 5-8, 11-14 or 22-24) must be better describe in this section in order to improve the explanation of the state of art of this investigation in the literature.

3) Omit the first person plural in the manuscript. Use passive sentences.

4) Line 68: Put a point between "test" and "the biocompatibility".

5) Who is the supplier of the graphite?

6) The extrusion processes must be better explained. Were GO and PLLA incorporated in the same zone of the extruder? Were they mixed previously? Were any plasticizer used? What is the pression, screew diameter, screew rate and production rate in the extruder? Was the specific mechanical energy calculated?

7) What is the molecular weight of the used PLLA?

8) What do you talk about the mechanical properties of the nanobiocomposites? Were they evaluated?

9) Line 106; Four what? Rats?

10) How was the statitical tests performed? Please indicate it in the experimental section of the manuscript.

11) Figure 1: The colors of the legend is difficult to follow.

Author Response

(The authors gave the same response as above.)

Round 2

Reviewer 2 Report

The authors have revised and corrected all the reviewer's comments. Thus, they have improve the quality of this work, which can be published in the present form.